# Knowledge Gap in Understanding the Steroidogenic Acute Regulatory Protein Regulation in Steroidogenesis Following Exposure to Bisphenol A and Its Analogues

**DOI:** 10.3390/biomedicines10061281

**Published:** 2022-05-30

**Authors:** Nur Erysha Sabrina Jefferi, Asma’ ‘Afifah Shamhari, Zariyantey Abd Hamid, Siti Balkis Budin, Adam Muhammad Zackry Zulkifly, Fatin Norisha Roslan, Izatus Shima Taib

**Affiliations:** Centre of Diagnostic, Therapeutic and Investigative Studies, Faculty of Health Sciences, Universiti Kebangsaan Malaysia, Jalan Raja Muda Abdul Aziz, Kuala Lumpur 50300, Malaysia; p119420@siswa.ukm.edu.my (N.E.S.J.); p109998@siswa.ukm.edu.my (A.‘A.S.); zyantey@ukm.edu.my (Z.A.H.); balkis@ukm.edu.my (S.B.B.); a168568@siswa.ukm.edu.my (A.M.Z.Z.); a168328@siswa.ukm.edu.my (F.N.R.)

**Keywords:** BPA analogues, cAMP, PKA, MAPK, ERK, Ca^2+^, Nur77, testicular steroidogenesis

## Abstract

The use of bisphenols has become extremely common in our daily lives. Due to the extensive toxic effects of Bisphenol A (BPA), the industry has replaced this endocrine-disrupting chemical (EDC) with its analogues, which have been proven to decrease testosterone levels via several mechanisms, including targeting the steroidogenic acute regulatory (StAR) protein. However, when exposed to BPA and its analogues, the specific mechanism that emerges to target StAR protein regulations remains uncertain. Hence, this review discusses the effects of BPA and its analogues in StAR protein regulation by targeting cAMP-PKA, PLC-PKC, EGFR-MAPK/ERK and Ca^2+^-Nur77. BPA and its analogues mainly lead to decreased LH in blood and increased ERK expression and Ca^2+^ influx, with no relationship with the StAR protein regulation in testicular steroidogenesis. Furthermore, the involvement of the cAMP-PKA, PLC-PKC, and Nur77 molecules in StAR regulation in Leydig cells exposed to BPA and its analogues remains questionable. In conclusion, although BPA and its analogues have been found to disrupt the StAR protein, the evidence in connecting the signaling pathways with the StAR regulations in testicular steroidogenesis is still lacking, and more research is needed to draw a solid conclusion.

## 1. Introduction

Bisphenols (BPs) are a class of chemical compounds composed of two phenolic rings connected by a carbon bridge or other chemical structures. BPs are frequently used in the production of polycarbonates and epoxy resins [1]. Bisphenol A (BPA) is the most frequently used BP that was introduced to the plastics industry in the 1950s. It is primarily used to make water bottles, shatterproof windows, and epoxy resins that coat canned food, bottle caps, and pipes. BPA has also been detected in baby bottles, and due to its toxic effects, many scientists are concerned about its hazardous impact on children’s development. Studies have indicated that BPA has detrimental effects on human health, including disruption of steroid hormone synthesis, mammary gland development, and changes in obesity-related parameters [2]. Moreover, BPA has been linked to many systemic diseases, such as cardiovascular disease, diabetes, and liver and renal diseases [3,4,5]. Therefore, the Centers for Disease Control and Prevention (CDC) conducted a survey, the National Health and Nutrition Examination Survey (NHANES III), and found detectable levels of BPA in 93% of 2517 urine samples in a cohort, ranging from six-year-old children to the elderly, in the United States [6]. Furthermore, in July 2012, BPA in many products, especially in baby products and packaging, was banned by the FDA. Consequently, developing a BPA alternative for industrial applications is critical. 

Some of the BPA alternatives are Bisphenol B (BPB), Bisphenol F (BPF), Bisphenol S (BPS), Bisphenol AP (BPAP), Bisphenol AF (BPAF), Bisphenol C (BPC), and Bisphenol Z (BPZ), as shown in Table 1. All of these analogues have almost the same structure as BPA, and the characteristic that distinguishes them from each other is the positioning of their respective functional groups. As their structure is almost identical to the BPA, these alternatives have been considered as a substitute for BPA in the manufacturing industry. The chemical structure of BPA and its analogues, as well as their applications and detection matrices, are summarized in Table 1. Most BP analogues are being used to replace BPA in the polycarbonate plastics and epoxy resin industries. BPB is utilized in a wide range of food-contact coating products, resulting in the presence of this chemical in the food in question. BPF is more suited to heavy industries that build solid or high systems [7]. BPS is used for the cleaning products industry, for instance, as a fastening agent for washing machines [8]. BPAF is primarily employed in producing specialized fluorelastomers as a crosslinker [9]. Meanwhile, BPC is commonly used in the production of fire-resistant polymers. Due to its widespread usage, these BPA analogues can be detected in most biological and environmental samples, such as air, water, and soil. These analogues can also be detected in food products, especially food stored in cans and plastics. These BPA analogues appear to have the potential to become global food contaminants and environmental pollutants in the near future. Furthermore, BPA analogues may have similar detrimental, physiological effects on organisms. 

BPA and its analogues are being classified as an endocrine-disruptor chemical (EDC) that acts as a systemic toxicant and has an impact particularly on the development of the reproductive system [10]. EDCs are substances found in the environment, food sources, personal care products, and manufactured products that disrupt the endocrine system’s normal function. Several studies have demonstrated that exposure to EDCs, such as BPs, during the prenatal and neonatal stages of development can cause harm and have a significant impact on the normal development of the reproductive system and functions in both males and females in adulthood. Furthermore, several studies have reported that BPs can cause male reproductive system toxicity by targeting the steroidogenesis process [11,12,13].

In general, Leydig cells and the adrenal gland are the sites of steroid hormone synthesis in mammalian cells. Steroidogenesis in Leydig cells is regulated primarily by the luteinizing hormone (LH) via several signaling pathways. It also involves a complex mechanism requiring the activation of several hormones and signaling pathways, including hydroxysteroid dehydrogenases (HSD) and cytochrome P450 enzymes [14]. In several studies, BPs have been shown to disrupt steroidogenesis by targeting the HSD and cytochrome P450 genes and proteins, which are either upregulated or downregulated as a consequence [11]. Cholesterol is a precursor in promoting testosterone synthesis, and steroidogenic acute regulatory (StAR) protein is vital in transporting cholesterol to the inner membrane of mitochondria in Leydig cells. StAR is regulated by three signaling pathways: (1) cyclic adenosine monophosphate (cAMP)–protein kinase A (PKA) (cAMP-PKA) and phospho-lipase C (PLC)-protein kinase C (PKC)(PLC-PKC) pathways; (2) epidermal growth factor receptor (EGFR)-induced mitogen-activated protein kinase (MAPK)/extracellular signal-regulated kinases (ERK)(MPAK/ERK); and (3) calcium ion (Ca^2+^) signaling and Nur77 transcription factor expression [15]. To the best of our knowledge, there are limited studies on the effects of BPA and its analogues in regulating the StAR gene and protein expression via these three signaling pathways. Hence, this review was carried out to understand a gap in the knowledge regarding these three signaling pathways, cAMP-PKA and PLC-PKC, EGFR-MAPK/ERK, and Ca^2+^-Nur77, in regulating the StAR gene and protein expression in case of exposure to BPA and its analogues.

## 2. Regulation of StAR in Normal Physiology 

Testicular steroidogenesis is a complex biological process in Leydig cells that entails the transformation of cholesterol to steroid hormones. The transportation of cholesterol from the outer to inner mitochondrial membrane in Leydig cells is mediated by the StAR protein, which plays a crucial role in steroidogenesis. The protein is produced as an active cytosolic protein in the form of a 37-kDa precursor containing an *N*-terminal mitochondria-targeting sequence that acts on the outer membrane of the mitochondria. This StAR protein migrates to the outer mitochondrial membrane after hormonal stimulation during steroidogenesis [16]. This 37-kDa StAR protein is then cleaved to a 30-kDa form that is phosphorylated to allow the transportation of the mass of cholesterol through the inner mitochondrial membrane [17]. The function of the StAR protein on the outer membrane of mitochondria is to desorb the cholesterol, pushing it from the outer membrane of the mitochondria and allowing it to cross the mitochondrial intramembranous space in microdroplet form before being taken up by the inner membrane of mitochondria [18]. Cholesterol is converted to testosterone via a series of steroidogenic enzymatic reactions through the action of HSD and CYP450 enzymes. It is vital to remember that phosphorylation of the StAR protein is essential for it to function well in steroid biosynthesis in terms of cholesterol transportation. 

StAR activity is stimulated by the luteinizing hormone (LH) and the follicle-stimulating hormone (FSH) [19,20]. LH and FSH bind to the G-protein coupled receptors known as the luteinizing hormone receptor (LHR) and follicle-stimulating hormone receptor (FSHR), respectively, for their specific functions in steroidogenesis. Interestingly, the molecular processes regulating Leydig cells’ response to steroidogenic stimuli may be distinct from those influencing FSH-responsive cells, such as ovarian granulosa cells [21]. In general, the binding of LH to the cognate LHR leads to StAR protein activation in the mitochondrial membrane of Leydig cells for the cholesterol transportation needed for testosterone synthesis [22,23]. A high testosterone concentration in the circulation through a negative feedback loop triggers the hypothalamus pituitary to suppress LH secretion. This results in the reduction of LHR-mediated stimulation of Leydig cells and ultimately decreases testosterone production. Positive feedback mechanisms are regulated in the opposite direction. 

Phosphorylation-dependent events are required for acute stimulation of steroid biosynthesis in all steroidogenic tissues via the activation of protein kinases, including cAMP-dependent protein kinase (PKA), protein kinase C (PKC), calcium/calmodulin-dependent protein kinase, and mitogen-activated protein kinases (MAPKs) [24,25]. Upon the binding of LH on LHR, the G protein dissociates into two subunits, one of which interacts with and activates an enzyme called adenylate cyclase (AC); this converts adenosine triphosphate (ATP) to a second messenger called cyclic adenosine monophosphate (cAMP) in the cell. An increase in cAMP activates the protein kinase A (PKA) pathway, which subsequently results in the phosphorylation of several transcription factors, such as the cAMP response element modulator (CREM) protein, cAMP response element-binding (CREB) protein, and GATA-binding protein 4 (GATA4). These transcription factors acting cooperatively and their interactions with the majority, if not all, of the cis-regulatory regions appear to be involved in regulating StAR gene expression. The cAMP–PKA pathway is the most well-studied mechanism linked to adult Leydig cell activity [26]. In addition to the activation of the cAMP-PKA pathway, phospholipase C (PLC) is also activated by LH/LHR complexes. The PLC splits phosphatidylinositol 4,5-bisphosphate (PIP2) into diacylglycerol (DAG) and inositol 1,4,5-trisphosphate (IP3), resulting in the activation of the Ca^2+^ channel in the endoplasmic reticulum by IP3. In combination with DAG, Ca^2+^ stimulates PKC, which phosphorylates CREB, thereby promoting StAR gene expression [25]. 

While the cAMP-PKA cascade is the predominant signaling pathway in controlling LH-stimulated StAR expression and steroid production, it is well documented that MAPK-ERK signaling cascades also play a critical role in regulating these processes [27,28]. A growing body of evidence indicates that MAPK signaling cascades are activated by a diverse array of extracellular inputs, such as cAMP, PKA, and PKC, which are critical in regulating the steroidogenic response [24,25]. PLC and PKC activate a cascade of protein kinases, such as Ras and Raf, resulting in many MAPK-ERK signaling cascades that play a critical role in regulating StAR expression and steroid production in steroidogenic tissues. Moreover, the PKA-signaling pathways can activate transcription factors directly or indirectly, and both of these signaling pathways are involved in the MAPK-mediated regulation of steroidogenesis. This pathway is known as the cAMP-dependent pathway. 

Besides the cAMP-PKA and PLC-PKC cascade inducer, activation of the epidermal growth factor receptor (EGFR) by extracellular stimuli, including cytokines, growth factor, ligands, and transforming agents is also involved in StAR protein regulation [29,30]. This event is known as the cAMP-independent pathway. This initiates a cascade of protein kinases, including Ras/Raf and other highly similar kinases. These protein kinases then activate a variety of transcription factors, subsequently being phosphorylated. Phosphorylation of these transcription factors regulates the StAR gene and consequently steroid production. Besides its function in StAR protein and steroid synthesis, ERK1/2 also phosphorylates the StAR protein to promote cholesterol mobilization in the synthesis of steroid hormones [31,32].

Recent evidence has shown that StAR expression is regulated not only by cAMP but also by intracellular Ca^2+^ concentrations by increasing intracellular Ca^2+^ and stimulating the Ca^2+^–calmodulin (CaM) protein kinase pathway in Leydig cells [26]. Calcium-regulated signal transduction acts as a second messenger in the regulation of sex steroid hormone synthesis, specifically targeting the StAR enzyme in the early cascade. cAMP production also activates a calcium-signaling pathway by increasing intracellular Ca^2+^ concentrations in Leydig cells, subsequently stimulating the expression of the transcription factor Nur77 (NR4a1) [33,34,35]. Nur77 (NR4a1) also acts as a mediator of hormone-stimulated StAR expression. Figure 1 shows the three signaling pathways regulating StAR expression: (1) cAMP-PKA and PLC-PKC, (2) EGFR-MAPK/ERK, and (3) Ca^2+^-Nur77.

## 3. Effects of BPA and Its Analogues on the LH-LHR, cAMP-PKA and PLC-PKC Signaling Pathways

The involvement of the cAMP-PKA and PLC-PKC signaling pathways in testicular steroidogenesis has been studied extensively. The interaction between LH and its receptor LHR on the membrane of Leydig cells is also well documented as a precursor in initiating testosterone production via the cAMP–PKA pathway. Previous studies have shown that BPA and its analogues have various effects on the LH, as the LH level is either increased or decreased upon exposure to BPA and its analogues. Several studies have shown that exposure to BPA, either in vivo or in vitro, causes a decrease in the level of LH in the blood circulation [36,37,38,39,40,41,42,43,44]. Most previous studies had shown that when adult male rats were exposed to multiple dosages of BPA ranging from 25 mg/kg/bw to 200 mg/kg/bw via oral gavage or intra-peritoneal injection, their LH levels decreased [36,37,38,40,41,42,43,45]. Furthermore, chronic exposure to BPS in drinking water for 48 weeks also reduced the production of LH in male rats [44]. The reduction of LH in blood circulation is due to the involvement of BPA and its analogues as EDCs, which inhibits hypothalamus–pituitary–gonad (HPG) axis regulation [46]. On the other hand, BPA increased the levels of LH after 15 days of exposure via oral gavage in Long–Evans rats [47]. A lack of responsiveness in the negative feedback regulation by testosterone on the HPG axis may occur in BPA-treated rats, resulting in serum LH overload [47]. 

Interestingly, previous studies also found that exposure to BPA and its analogue BPAF increased the level of LH but decreased LHR expression [48,49]. Savchuk, Söder, and Svechnikov [48] found in their research that BPA at a concentration of 10 mM increased LH levels but decreased LHR expression in Leydig cells isolated from the C57BL/6j strain. Facilitated by BPA, increased LH levels and decreased LHR expressions are impacted by Leydig cells’ capacity to produce testosterone and estradiol [48]. Moreover, the protein expression of LHR deteriorated even when LH secretion increased substantially in BPAF-treated rats [49]. Increased LH levels in serum were shown to be associated with a significant decrease in StAR protein expression in the testes of rats treated with 200 mg/kg/bw of BPAF. An explanation for these observations is that the effect of increasing LH levels on StAR expression may be abolished by the lower LHR expression in Leydig cells [49]. In contrast, an increased LHR level was detected in BPF-treated male zebrafish (Danio rerio) at concentrations of 0.1 and 1 mg/L in the tank [39]. Even though the LHR upregulation increased, the BPF exposure decreased StAR gene expression, resulting in a decrease in testosterone production [39].

Exposure to BPA in vitro studies decreased cAMP expression [50,51]. Both studies used the same cell lines, which are mouse Leydig tumor cells (mLTC), pre-incubated with BPA for 48 h and 1 h at concentrations of 10−8 mol/L and 0.01 Μm–10 μM, respectively. The researchers hypothesized that BPA attenuates hCG-stimulated cAMP, which then reduces LHR-mediated signal transduction, thereby preventing LH from activating the AC. This eventually leads to the inhibition of the cAMP pathway in such a way that BPA also triggers a decrease in intracellular cAMP accumulation [50]. In contrast, rat Leydig cells exposed to BPA at a concentration of 0.1 nM increased the cAMP expression after 24 h of exposure [52]. Furthermore, Kim et al. [52] also found that BPA increased the phosphorylation of PKA in response to an increase in cAMP expression. Furthermore, this cAMP-PKA activation also increased CREB phosphorylation and might have triggered the MAPK-ERK signaling pathway [52]. To the best of our knowledge, only one study reported the effect of BPA on the cAMP-PKA signaling pathway in testicular steroidogenesis in Leydig cells. Several previous studies focused on the cAMP-PKA signaling cascade in myocardial, sperm, and granulosa cells [53,54,55,56,57,58,59]. Researchers have found that exposure to BPA activates oxidative stress via increased reactive oxygen species (ROS) production, leading to an increase in the cAMP-PKA signaling cascade. Moreover, no study reported the effects of BPA and its analogues on the PLC-PKC pathway involved in testicular steroidogenesis in Leydig cells. Table 2 shows the effects of BPA and its analogues on the LH, LHR, and cAMP-PKA signaling pathways in the male reproductive system.

## 4. Effects of BPA and Its Analogues on EGFR-MAPK-ERK Signaling Pathway

Physiologically, despite the cAMP-dependent system, an independent mechanism also plays a crucial role in regulating the StAR protein. Several studies have suggested that cytokines and growth factors can regulate the steroidogenic pathway of Leydig cells in a cAMP-independent manner via their ability to bind to EGFR [29,30]. The pituitary hormone is crucial for steroidogenesis, as proven in granulosa and theca cells treated with FSH and LH, respectively. Both hormones cause selective inhibition of MAPK-ERK1/2, resulting in the inhibition of StAR protein, progesterone, and androgen synthesis. However, in aging, ERK activation controls mitochondrial fusion and StAR localization in mitochondria results in steroidogenesis disturbance in Leydig cells. Therefore, the role of ERK1/2 in influencing steroidogenesis is still being explored and articulated, as it has been found to be inhibitory in some studies, while other investigations have shown a beneficial influence of MAPK activation on the synthesis of sex steroids [62,63]. 

Notably, BPA and its analogues disrupted this cAMP-independent pathway involving MAPK-ERK; however, there is insufficient evidence to link all the information with steroidogenesis. As previously stated, MAPK-ERK not only regulates StAR but also plays a role in cell proliferation, with many research reports on this effect. Exposure to BPA during GD12 until weaning at PND21 increased EGFR and MAPK expression; however, it did not cause any alteration in the testosterone levels of male pups. This is due to the increased number of Leydig cells that compensate for testosterone synthesis [64]. The cAMP-independent pathway is involved in StAR regulation and Leydig cell proliferation [64]. The same finding was also reported in an in vitro study performed in a dose-dependent manner. When TM3 Leydig cells were exposed to BPA for 24, 48, and 72 h, BPA increased the phosphorylation of ERK1/2 and Akt in Leydig cell proliferation [65]. Ok et al. [66] found that BPA increased the phosphorylation of ERK1/2 in vitro and in vivo. Exposure to BPA at a concentration of 100 µM for 24 h increased ERK 1/2 phosphorylation in TM3 Leydig cells. The phosphorylation of ERK1/2 was also increased in the testicular tissue of rats injected with 50 mg/kg/bw of BPA once a week [66]. 

Exposure to the BPA analogue BPB also increased the ERK1/2 phosphorylation in Leydig cells. Exposure to BPB at doses of 100 and 200 mg/kg disrupts Leydig cell maturation in late puberty by increasing Leydig cell proliferation via increased ERK1/2 expression. Interestingly, a significant reduction was found in testosterone levels after exposure to the same dose of BPB. However, the study did not report any connections between ERK1/2 expression with the regulation of StAR but only reported its proliferation effects in Leydig cells [67]. Expression of ERK1/2 phosphorylation was reported in rat testicular Leydig cell R2C when exposed to BPA for 30 min at various concentrations (0.1, 1, and 10 nM). Furthermore, the exposure also caused activation of alternative testicular steroidogenesis pathways, as proven by increased aromatase activity. These findings suggest that the activation of aromatase is correlated to the MAPK-ERK signaling pathway [50]. Exposure of pregnant rats to BPA at a dose ranging between 0.1 and 200 mg/kg/day from gestational day 14 (GD14) to birth (D0) resulted in increased ERK1 and Raf1 expression in the testicular tissue of male offspring at PND3, 21, and 60 [68]. Notably, Thuillier et al. [68] also found that ERK1 is more prominently expressed in Sertoli cells, while Raf1 is more detectable in Leydig cells, as proven by immunohistochemistry staining. Raf1 participates in the MAPK-ERK signaling cascade during testicular steroidogenesis via a cAMP-independent event. Despite the lack of evidence on the MAPK-ERK signaling pathway in regulating the StAR protein in testicular steroidogenesis, a study by Chu et al. [69] found that the placenta exposed to BPA influenced the ERK signaling pathway by increasing the phosphorylated ERK and resulted in steroidogenesis disturbance that was proven by a reduction in progesterone levels [69]. Nevertheless, the effects of BPA and its analogues in regulating the StAR protein via the MAPK-ERK signaling pathway remain unknown. Table 3 shows the effects of BPA and its analogues on the MAPK-ERK signaling pathway in the male reproductive system.

## 5. Effects of BPA and Its Analogues on the Ca^2+^ Signaling and the Involvement of Nur77 Transcription Factor

Another signaling pathway is also reported to be involved in the upregulation of StAR, which is the calcium-regulated signal transduction. Calcium-regulated signal transduction serves as a second messenger involved in regulating sex steroid hormone synthesis, particularly targeting the early cascade, of the StAR enzyme [70]. When LH binds to its receptor, it increases intracellular cAMP levels, followed by an increase in the concentration of intracellular Ca^2+^. There are, however, other indications that LH increases in the concentration of intracellular Ca^2+^ and that its action is dependent on the presence of calcium in the extracellular medium (known as extracellular Ca^2+^), whose influx across the plasma membrane is required for the steroidogenic process to occur. The elevated influx of Ca^2+^ in Leydig cells stimulates the expression of the transcription factor Nur77. Nur77 plays a mediating role in stimulating StAR expression. Recently, there has been abundant research that correlates the expression of StAR protein with Nur77 [33,34,35]. The knockout and overexpression of Nur77 clearly showed no StAR expression or overactivity, respectively, in Leydig cells. 

In previous ex vivo studies, exposure to BPA and its analogue TBPA showed a significant increase of Ca^2+^ [71,72,73]. Batista–Silva et al. [71], in an ex vivo study, found that BPA exposure at concentrations of 10 pM, 10 nM, and 10 µM for 30 min stimulated Ca^2+^ influx. According to their findings, BPA-induced Ca^2+^ influx occurred via the PKC pathway, as it was proven that exposure to PKC inhibitor abolished the stimulatory effect of BPA on Ca^2+^ influx. Furthermore, the researchers also found that IP3R was involved in inducing Ca^2+^ influx in the PKC pathway with BPA exposure [71]. The same finding was also noted in a study conducted by Gonçalves et al. [72]. Testes isolated from 30-day-old male Wistar rats exposed to BPA for 5 min at concentrations of 0.1 pM, 1 pM, and 10 nM showed increased Ca^2+^ influx. Treatment with PLC inhibitors terminated the effects of BPA-induced Ca^2+^ influx in the testis [72]. Furthermore, exposure to BPA and its analogue TBBPA at a concentration of 10 nM increased the Ca^2+^ influx in immature boar testes after incubation for 48 h [73]. 

An increased level of Ca^2+^ triggers Nur77 expression in Leydig cells. Based on previous findings, BPA and its analogue BADGE.2H2O showed an increased level of Nur77 gene expression in the mLTC line (K28) after 24 h of exposure [74,75]. Based on Song et al.’s [75] research, exposure to BPA in vitro induces Nur77 mRNA expression in a dose-dependent and time-dependent manner. However, their study showed no involvement of PKC in Nur77 gene expression, because no changes were recorded after the PKC inhibitor was inserted. Moreover, Ahn et al. [74] found that exposure to BADGE.2H20 increased Nur77 mRNA expression; however, the expression of StAR was found to decrease. Increased Nur77 expression may involve inflammatory response and apoptosis but not StAR transcription regulation [76]. Table 4 shows the effects of BPA and its analogues on Ca^2+^ influx and Nur77 transcription factors in the male reproductive system.

## 6. Effect of BPA and Its Analogues on StAR Protein Expression

The cAMP-dependent and independent mechanisms play an important role in regulating StAR expression in testicular steroidogenesis. StAR mediates the transportation of cholesterol, which is assumed to be the rate-limiting step in steroid hormone production. Therefore, disruption of StAR, either gene or protein expressions, influences the transportation of cholesterol, which is the precursor substance involved in testosterone synthesis. Past studies have indicated varied effects of BPA and its analogues: there is either increased, decreased, or no effect on the StAR genes or protein expressions. The differences in the StAR gene and protein expressions may be due to the different types and ages of the animal models used in the studies on exposure to BPA and its analogues.

Reduced StAR protein expression was found in Leydig cells as well as in the testicular tissue of the mice that were intraperitoneally injected with BPA. Furthermore, serum testosterone levels were also found to be significantly lower in BPA-treated mice at a dose of 20 mg/kg after 7 days of exposure [77]. StAR gene expression was found to be downregulated in embryo testes when pregnant mice were given drinking water contaminated with 10 µM of BPA in combination with radiation exposure [78]. The StAR protein was significantly reduced after 24 h of exposure to 70 mM of BPAF in the mouse Leydig cell tumor cell line (mLTC-1). However, after being treated with 22R-hydroxycholesterol, the researcher concludes that BPA did not cause alterations in StAR protein but inhibited SR-B1, a protein that is responsible for transporting plasma cholesterol to the steroidogenic tissues and may be a molecular target in this steroidogenic process of mLTC-1 exposed to BPAF [49].

In contrast, it has been revealed in several studies that BPA and its analogues increased StAR gene and protein expression [77,79,80,81,82]. Expression of the StAR gene was reported to be elevated significantly in a BPA-exposed group of sexually immature male medaka fish. The researcher suggested that the effects of BPA may be different when exposed during the active reproductive state [79]. An ex vivo study found that BPA at the concentrations of 10 µM and 100 µM causes a significant increase in StAR gene expression in fetus and adult testes exposed for 5 days [83]. A study was carried out by Chen et al. [80] using the ethane dimethane sulfonate (EDS)-induced Leydig cell regeneration model to assess the impact of BPA exposure on the differentiation of stem cells in vivo and in vitro, targeting the specific genes LHR and StAR. BPA was found to increase testosterone levels and Leydig cell-specific genes (LHR, StAR) and their protein expression levels in vivo. Similar findings were also observed in an in vitro study; however, StAR gene and protein expression remained unchanged. Interestingly, germ cells exposed to BPA, BPF, and BPS and a combination of all bisphenols for 24, 48, and 72 h increased StAR gene expression. The researchers concluded that BPS and BPF appear to act similarly to BPA, proven by the increase in StAR gene expression, and thus may represent comparable reproductive concerns [82].

In a comparison between BPA, BPE, and BPS, the relative mRNA expression of StAR was found to increase upon BPS exposure at doses of 0.5 and 50 g/kg/day at prenatal age [81]. There were no BPB-induced changes in StAR gene and protein expressions in relation to Leydig cell maturation in late-puberty rats. BPB alters Leydig cell maturation by increasing the number of Leydig cells, thus reducing testosterone synthesis and downregulating steroidogenesis-related gene expression; however, it does not involve the StAR-related genes [67]. Moreover, the ability of BPA analogues to adversely affect StAR gene expression was reported. Silico docking studies exhibited that BPS, in binding with the StAR protein, may influence the transportation of cholesterol into mitochondria, which in turn influences steroidogenesis [58]. Logically, an increase in StAR gene and protein expression may increase testosterone production. However, most studies have stated that an increase in StAR causes testicular steroidogenesis defects. Even though StAR gene and protein expressions were increased upon exposure to BPA and its analogues, such StAR formation entails an inactive form that is not phosphorylated. Therefore, cholesterol transportation will be affected, resulting in a decrease in testosterone synthesis. Table 5 shows the effects of BPA and its analogues on the StAR protein in testicular steroidogenesis.

## 7. Discussion

BPA and its analogues were found to disrupt StAR protein and gene expression; however, the data linking the signaling pathways to the regulation of StAR in testicular steroidogenesis remains insufficient. The StAR protein and genes are regulated via several pathways, including cAMP-PKA and PLC-PKC, EGFR-MAPK/ERK, and Ca^2+^-Nur77. Several studies have found that BPA and its analogues reduce the level of LH in blood circulation [36,37,38,39,40,41,42,43,44]. However, studies carried out by Bai et al. [60] and Akingbemi et al. [47] reported an increase in LH levels in blood circulation after BPA exposure. Notably, exposure to BPA and BPAF has been reported to cause an increase in LH level and a decrease in LHR expression in Leydig cells [48,49]. On the other hand, LHR expression was upregulated in the testes of male zebrafish treated with BPF, as indicated only by Yang et al. [39]. A previous study also showed that exposure to BPA and its analogues leads to HPG axis disturbances, resulting in contradictory levels between LH and LHR [11,47]. Moreover, it was revealed that decreased expression of LHR reduced the capacity of LH to activate the signaling pathways. This mechanism reduces StAR gene expression and reduces testosterone synthesis. Therefore, the mechanism and roles of LH in StAR production and regulation in Leydig cells under exposure to BPA and its analogues requires attention and further investigation. When LH binds to LHR, the cell converts ATP to cAMP. An increase in cAMP stimulates the PKA pathway, leading to the phosphorylation of many transcription factors required for the StAR protein to function well. BPA adversely affects R2C rat Leydig cells by increasing the level of cAMP. Interestingly, upon exposure to BPA, cAMP production in pre-incubated mLTC-1 decreases even after human LH is added to the cells [50]. This shows that BPA is capable of inhibiting cAMP activation upon LH stimulation. To the best of our knowledge, there are scarce findings regarding the effects of BPA and its analogues on PKA stimulation and transcription factors involved in regulating StAR protein in Leydig cells. In addition, there is no available evidence regarding the impacts of BPA and its analogues on PLC-PKC pathway activation in Leydig cells. Thus, this could be an opportunity for researchers to explore in depth EDCs’ effects on this potential targeting protein, PKC.

Furthermore, the MAPK-ERK signaling pathway is crucial in activating StAR protein expression. An increase in ERK phosphorylation through the MAPK-ERK signaling pathway was found in animal models that were exposed to BPA and its analogues [52,64,65,66,67,68,69]. However, there is no clear evidence on how BPA and its analogues affect the MAPK-ERK signaling pathway in regulating StAR protein expression. MAPK phosphatase in steroidogenic cells plays a role in turning off hormonal signaling in ERK-dependent activities, such as steroid production and cell proliferation. Moreover, a comprehensive list of previous studies illustrated that BPA and its analogues induce the modification of this signaling pathway by triggering Leydig cell proliferation. These findings possibly indicated that BPA exposure may alter the MAPK-ERK signaling pathway by switching off its normal function of inducing StAR phosphorylation and thus causing excessive cell proliferation, which can lead to tumor cell formation.

Calcium-regulated signal transduction acts as a second messenger in the regulation of sex steroid hormone production, specifically targeting the StAR protein. BPA and its analogues can also interfere with the Ca^2+^ transduction system by increasing the influx and concentration of Ca^2+^ and Nur77 expression in ex vivo and in vitro models. From the collected data, Ca^2+^ concentration after exposure to BPA and its analogue is shown to consistently increase. Ca^2+^ plays a critical role in stimulating StAR via its StAR protein transcription ability. However, an increase in Ca^2+^ upon BPA exposure may not trigger StAR transcription but is involved in oxidative stress in the mitochondria. The increased concentration of Ca^2+^ can stimulate the formation of ROS in the mitochondria during respiration, leading to oxidative stress [84]. Ca^2+^ in Leydig cells can stimulate Nur77 expression, which has a role in the stimulation of StAR protein expression and transcription. In addition, all the findings showed an increased in Nur77 expression after BPA exposure. Theoretically, the increase of Nur77 expression will increase the inactive form of the StAR protein. Usually, the StAR protein should be activated for it to function via phosphorylation. Hence, exposure to BPA and its analogues may only increase the inactivated form of the StAR protein, leading to a reduction in its function for cholesterol mobilization. Furthermore, the elevated Nur77 expression has been reported to be involved in inflammation. Such Nur77 may switch its role from being involved in StAR expression and transcription to the inflammatory response [76].

Previously published research suggested that BPA and its analogues have a variable influence on StAR gene and protein expression—either increasing, decreasing, or having no effect. These may be due to the differences in the dose of exposure, the route, and the organisms used in the respective studies. Even though the results vary, the changes in the StAR protein can lead to testicular steroidogenesis defects. Therefore, we suggest that BPA and its analogues can affect StAR protein regulations in steroidogenesis via several pathways, such as activation of cAMP-PKA, EGFR-MAPK/ERK, and Ca^2+^-Nur77 signaling pathways, which then ultimately decrease the StAR protein or its functionality.

This review article is more focused on understanding the knowledge gap in the three signaling pathways (i.e., cAMP-PKA, MAPK-ERK, and Ca^2+^-Nur77) regulating the StAR protein in testicular steroidogenesis upon exposure to BPA and its analogues. However, the findings from our literature search showed that researchers reported only on individual signaling pathways, such as cAMP-PKA, MAPK-ERK, and Ca^2+^-Nur77, leading to a lack of evidence in linking these pathways with the StAR protein in Leydig cells. We also found that other researchers conducted more studies on other cells, such as myocytes, sperm, and granulosa cells, in relation to these three signaling pathways. Most of our findings showed decreased LH in blood circulation and increased ERK expression and Ca^2+^ influx; however, previous researchers did not describe in detail the relationship between these molecules and StAR protein expression. BPA and its analogues decreased LH in blood circulation, resulting in decreased testosterone synthesis due to the downregulation of LHR. Moreover, BPA and its analogues may also activate the cAMP-PKA pathway, thus activating the MAPK-ERK1/2 cascade. Exposure to BPA may also increase the expression of EGFR in the Leydig cell membrane, which also triggers MAPK-ERK1/2 cascade activation. Most of the studies reported that activation of MAPK-ERK1/2 upon exposure to BPA and its analogues resulted in an increase in Leydig cell proliferation instead of increasing the phosphorylation of StAR. BPA and its analogues also increased Ca^2+^ influx and Nur77 expression in Leydig cells, which may be involved in oxidative stress and inflammatory response, respectively, but not involved in regulating the transcription of the StAR protein. Therefore, we postulate the possible effects of BPA and its analogues in regulating StAR gene and protein expressions via these three signaling pathways in Leydig cells, as shown in Figure 2.

BPA and its analogues disturb the StAR gene and protein regulations via these three signaling pathways, ultimately having a negative impact on human health. The StAR regulation disturbance after exposure to BPA and its analogues, targeting these three signaling pathways can downregulate the phosphorylated-StAR, also known as activated StAR. The inactive form of the StAR protein will decrease testosterone production in Leydig cells and activate aromatase activity. Increased activation of aromatase activity may increase the estradiol in the male reproductive system, which ultimately increases prostate cancer risk. Therefore, targeting these three signaling pathways in regulating the StAR protein can also benefit therapeutic approaches in alleviating the adverse effects of these EDCs.

## 8. Conclusions

In conclusion, BPA and its analogues represent an additional risk factor to consider due to their disruption of StAR gene and protein expressions during testicular steroidogenesis. The findings of the experimental studies mostly showed that BPA and its analogues can disrupt the LH levels and LHR expression in the Leydig cell membrane, leading to the activation of cAMP-PKA, MAPK-ERK1/2, and Ca^2+^ influx; however, there is a lack of evidence connecting all three signaling pathways with StAR regulation. Therefore, more research is required to establish the toxic effects of these EDCs in large populations and their molecular mechanism targeting the cAMP-PKA, PLC-PKC, MAPK-ERK1/2, Ca^2+^ influx, and Nur77 transcription factor to gain a better understanding of StAR regulation in the Leydig cell population. Understanding the knowledge gap with regard to these three signaling pathways (i.e., cAMP-PKA, MAPK-ERK, Ca^2+^-Nur77) in regulating the StAR protein can be used to determine target molecules when developing a drug for the regulation of testicular steroidogenesis. Furthermore, these data can also be used as additional information related to BPA and its analogues’ toxicity in the male reproductive system, specifically concerning testicular steroidogenesis.

## Figures and Tables

**Figure 1 biomedicines-10-01281-f001:**
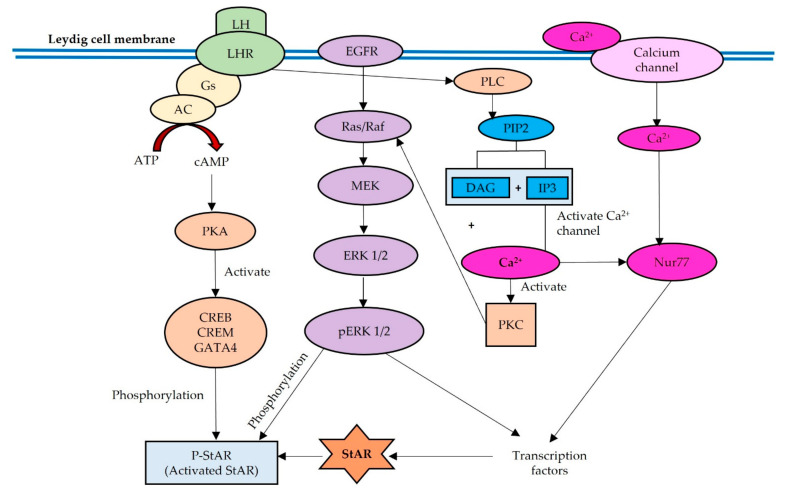
Three signaling pathways in regulating StAR expression: cAMP-PKA and PLC-PKC, EGFR-MAPK/ERK, and Ca^2+^-Nur77.

**Figure 2 biomedicines-10-01281-f002:**
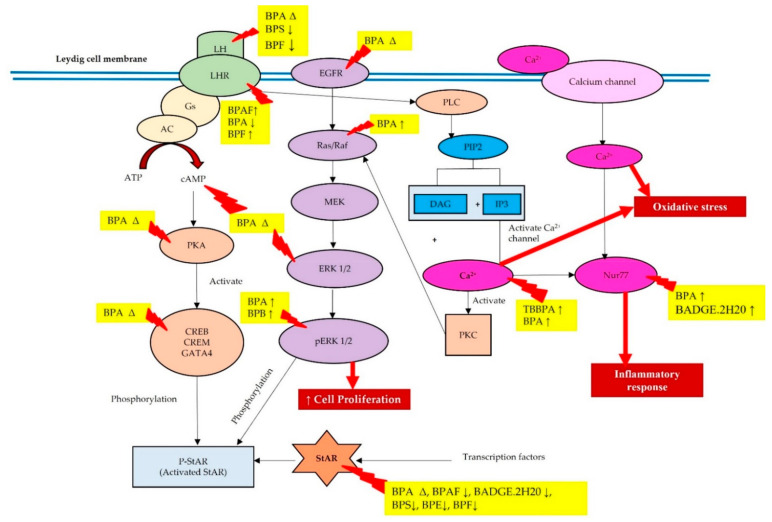
The effects of BPA and its analogues targeting the three signaling pathways in regulating the StAR gene and protein expressions. Abbreviations: ↑, increase; ↓, decrease; ∆, changes (can be increase or decrease).

**Table 1 biomedicines-10-01281-t001:** The chemical structure of BPA and its analogues and detection matrices.

Bisphenol	Chemical Structure	Usage	Detection Matrices
Bisphenol A	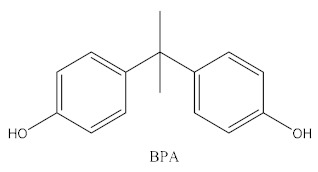	Hard plastic items (baby bottles, reusable water bottles, food containers, pitchers, tableware and other storage containers); polycarbonate plastic (eyeglass lenses, CDs, DVDs, computers, appliances, sports safety equipment); epoxy resin linings coat the inside of metal products (foo d cans, bottle tops and water supply pipes).	Air, dust, water, blood, urine, sediment, food, municipal sewage sludge
Bisphenol B	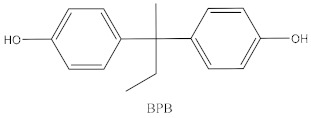	Food-contact coatings, polymers	Food, dust, sediment, blood, urine
Bisphenol F	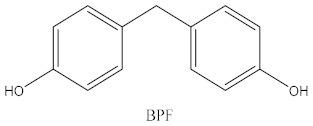	Epoxy resins, polycarbonates (lining of solid/high built systems); thermal receipt	Food, dust, sediment, receipts, urine, PCP, municipal sewage sludge
Bisphenol S	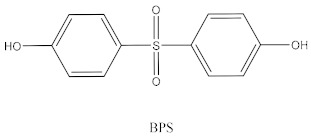	Wash fastening agent, electroplating solvent, thermal receipt papers	Blood, food, dust, sediment, receipts, urine
Bisphenol AP	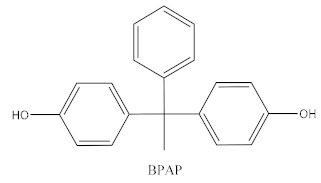	Polycarbonates, epoxy resins, polyarylates, polyethers, polyetherimides, polyphenylene ethers, copolymers	Food, dust, sediment, receipts
Bisphenol AF	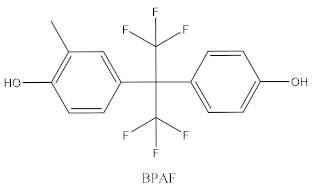	Crosslinker (specialty fluoro-elastomers synthesis)	Food, dust, sediment, municipal sewage sludge
Bisphenol C	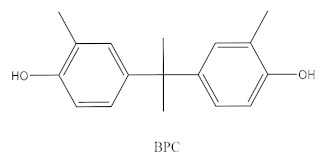	Production of fire-resistant polymers	Receipts

**Table 2 biomedicines-10-01281-t002:** The effects of BPA and its analogues on the LH, LHR and cAMP-PKA signaling pathways in the male reproductive system. Abbreviations: ↑ Increase; ↓ Decrease.

Type of Bisphenol	Purity (Manufacturer)	Dose (Route)	Animal Cells	Duration of Exposure	Findings	Author
BPA	Sigma–Aldrich	0.01 μM– 10 μM	Mouse Leydig Tumor Cells (mLTC)	1 h of preincubation	100 μM: ↓ cAMP	[50]
BPS	Sigma–Aldrich	(0, 0.5, 5, and 50 μg/L BPS mg/kg body weight/day) of BPS	Weaning Sprague–Dawley rats at postnatal day 22 (PND)	48 weeks	↓ LH	[44]
BPAF	(99%) Tokyo Chemical Industry	0, 2, 10, 50 and 200 mg/kg/day	Male Sprague–Dawley rats aged 7 weeks	14 days	200 mg/kg: ↓ LHR ↑ LH, FSH	[49]
BPA	Sigma–Aldrich	low (2.4 or 10 g/kg/D BPA) and high (100 or 200 mg/kg/d BPA) doses 0, 0.01, 0.1, 1, 10, 100, and 1000 nm BPA	Long–Evans strain of rat (Charles River, Wilmington, MA, USA) Adult Leydig cells obtained from 90-day-old rats with 0, 0.01, 0.1, 1, 10, 100, and 1000 nm BPA for 18 h	15 days of oral gavage 18 h incubation for adult Leydig cell	↑ LH	[47]
BPA	Sigma–Aldrich (natural exposure from polycarbonate cage)	10^−8^ mol/liter	Leydig tumor cells (mLTC-1 cells)	preincubation of mLTC-1 cells for 48 h	↓ cAMP	[51]
BPA	Sigma–Aldrich	0.1 nM	rat Leydig R2C cells	24 h	↑ CREB ↑ cAMP ↑ PKA phosphorylation	[52]
BPA	Sigma–Aldrich	10 mM	CBA/Lac, C57BL/6j, BALB/c and 129S2 mouse strains	17 h	↓ LHR ↑ LH	[48]
BPA	Sigma–Aldrich	2 mg/kg/bw (s.c)	Offspring male Sprague Dawley	Perinatal exposure day 10 of gestation until day 7 of lactation	↑ LH	[60]
BPF	J&K Scientific Ltd.	0.1 and 1 mg/Lin aquarium water	Male Zebrafish	21 days	↑ LHR	[39]
BPA	Sigma-Aldrich	5 or 25 mg/kg/bw (oral gavage)	Adult male Wistar rats	40 days	↓ LH	[36]
BPA	Sigma–Aldrich	50 mg/kg/bw (oral gavage)	Adult male Wistar rats	14 days	↓ LH	[40]
BPA	Sigma-Aldrich	50 mg/kg/bw (oral gavage)	Adult male Wistar rats	30 days	↓ LH	[42]
BPA	Sigma-Aldrich	200 mg/kg (oral gavage)	Adult male SD rat	42 days	↓ LH	[37]
BPA	-	25 mg/kg/bw (i.p.)	Adult male SD rats	Alternate day for 30 days	↓ LH	[43]
BPA	Gracia chengdu chemical technology co	200 mg/kg (oral gavage)	Adult male SD rats	28 days	↓ LH	[38]
BPF	Santa Cruz Biotechnologies	1, 5, 25, 50, and 100 mg/kg/bw (Oral gavage)	Adult male SD rats	28 days	↓ LH	[61]

**Table 3 biomedicines-10-01281-t003:** The effects of BPA and its analogues on the EGFR-MAPK-ERK signaling pathway in the male reproductive system. Abbreviations: ↑ Increase; ↓ Decrease.

Type of Bisphenol	Purity (Manufacturer)	Dose (Route)	Animal/ Cells	Duration of Exposure	Findings	Author
BPA	Sigma-Aldrich	2.5 or 25 ug/kg bw (oral gavage)	Long-Evans (LC)	GD 12 through weaning at PND 21 Assessment of Leydig cells differentiation at 21, 35, and 90 days of age of male pups	↑ EGFR ↑ MAPK	[64]
BPA	Sigma-Aldrich	10^−8^ to 10^−3^ M	LC TM3	24, 48, or 72 h	↑ phosphorylation of ERK1/2 and Akt	[65]
BPB	(>98%) Tokyo Chemical Industry	10, 100, and 200 mg/kg/bw (oral gavage)	Male Sprague-Dawley (35 days old) rats	21 days	↑ the phosphorylation of AKT1, AKT2, and ERK1/2 at 100 and 200 mg/kg ↓ Testosterone	[67]
BPA	Sigma-Aldrich	(0–200 µM)	JEG-3, a human choriocarcinoma cell line	24 h (expose for 48, 72, 96 h)	↑ phosphorylated ERK ↓ progesterone	[69]
BPA	Sigma-Aldrich	0.1, 1, 10 nM	Rat testicular Leydig R2C	30 min	↑ phosphorylation of ERK1/2 ↑ aromatase activity	[52]
BPA	Sigma-Aldrich	100 μM in 0.1% DMSO 50 mg/kg body weight/week	TM3 LC Male SD rats (i.p)	24 h 7 days	↑ phosphorylation of ERK1/2 ↑ phosphorylation of ERK1/2	[66]
BPA	(>98%) Sigma-Aldrich	0.1 to 200 mg/kg/day	Pregnant SD Male offspring at postnatal day (PND) 3, 21, or 60	(GD14) to birth (D0)	↑ ERK1, p-ERK1 (More prominent in Sertoli cell) ↑ Raf1 (more prominent in Leydig cell)	[68]

**Table 4 biomedicines-10-01281-t004:** The effects of BPA and its analogues on Ca^2+^ influx and Nur77 transcription factor in the male reproductive system. Abbreviations: ↑ Increase; ↓ Decrease.

Type of Bisphenol	Purity (Manufacturer)	Dosage (Route)	Animal/ Cells	Duration of Exposure	Findings	Author
BPA	Sigma–Aldrich	10 pM, 10 nM & 10 µM	*D. rerio* (Ex vivo)	5, 10, 15, 20, and 30 min incubation	5, 10, 15, 20: No effect 30 min: BPA treatment at 10 pM and 10 nM stimulated Ca^2+^ influx	[71]
BPA	Sigma–Aldrich	0.1 pM, 1 pM & 10 nM	Thirty-day-old male Wistar rats of testes (Ex vivo)	5 min incubation	BPA induces Ca^2+^ influx involved PKC activation in rat PLC inhibitors terminate the effect of BPA-induced Ca^2+^ influx.	[72]
BPA TBBPA	Sigma-Aldrich Santa Cruz	10 nM	immature boar testis (Ex vivo)	48 h	BPA and TBBPA ↑ (*p* < 0.01) of Ca^2+^ concentration	[73]
BPA & BADGE.2H2O	Sigma-Aldrich	1 µM	K28 mouse Leydig tumor cell line	24 h	BADGE.2H2O and BPA treatment ↑ Nur77 mRNA expression BADGE.2H2O treatment decrease the StAR expression	[74]
BPA	Sigma-Aldrich	1 μM	K28 mouse Leydig tumor cell line	24 h	BPA specifically induces Nur77 gene expression in a time- and dose-dependent manner No changes of Nur77 after PKC inhibitor. BPA ↑ Nur77 gene promoter activity and its transactivation	[75]

**Table 5 biomedicines-10-01281-t005:** The effects of BPA and its analogues on the StAR protein in testicular steroidogenesis. Abbreviations: ↑ Increase; ↓ Decrease.

Type of Bisphenol	Purity (Manufacturer)	Dose (Route)	Animal/ Cells	Duration of Exposure	Findings	Author
BPA	Sigma-Aldrich	in vivo: 20 mg/kg (i.p.) ex-vivo: 0, 5, 10, 20, 40, and 80 mM of BPA	C57BL/6J male wildtype (WT) mice Primary LCs	7 days 24 h	↓ StAR ↓ serum testosterone level	[77]
BPA	>99% (Merck)	10 µM BPA (diluted in 0.1% ethanol) in drinking water	Pregnant mice (testes embryo)	10.5 days post-coïtum (dpc) to 18.5 dpc.	Combination of radiation and BPA: ↓ gene expression (combination of StAR, Hsd3b1 and Hsd17b3)	[78]
BPA	>99% (Sigma-Aldrich)	In vivo: 10, 100 or 1000 pmol/testis (intratesticular injection) In vitro: 1, 10, 100, and 1000 nmol/L	SD (60 days)	In vivo: Post-EDS days 7–28 for 21 days	In vivo: 100 and 1000 pmol/testis of BPA from post-EDS day 14–28: ↑ serum testosterone ↑ Leydig cell-specific gene (Lhr, Star and their protein expression levels. No alteration: LH, FSH & proliferative capacity of Leydig cells in vivo. In vitro: 100 nmol/L stimulated the differentiation of stem Leydig cells by: ↑ testosterone levels up-regulating Leydig cell-specific Lhr gene and proteins but did not affect their proliferation.	[80]
BPAF	>99% (Tokyo Chemical Industry)	0, 0.1, 1, 10, 30, 50, and 70 mM	mLTC-1 cell	24 h	↓ StAR after exposure to 70 mM BPAF. No alteration of StAR after treated with 22R-hydroxycholesterol	[49]
BPB	>98% (Tokyo Chemical Industry)	10, 100 and 200 mg/kg/day (Oral gavage)	Male SD	(PND) 35 to PND 56	No effects of expression of StAR	[67]
BPA	NA	BPA (10 μg/L) (tank)	Male medaka fish	from 8 h post-fertilization (as embryos) to adulthood 50 days post fertilization (dpf)	↑ expression StAR gene pattern	[79]
BPA, BPF and BPS	Sigma Aldrich	BPA: 10^−8^ M BPF: 10^−8^ M BPS: 10^−8^ M	Germ cell line	24, 48, 72 h	↑ StAR gene expression at 24, 48 and 72 h exposure	[82]
BPA, BPE and BPS	Sigma Aldrich	0.5 or 50 mg/kg/day	CD-1 mice	GD7 to birth	↑ relative mRNA expression of Star in BPS	[81]
BPA	Sigma Aldrich	10 µM 100 µM	Fetal testis	5 days	↑ StAR gene expression in BPA-treated fetal & BPA exposed testes	[83]
BPS	98% (Sigma Aldrich)	50 μg/L (drinking water)	Male Wistar rats on the post-natal day (PND) 21	10 weeks	In silico docking: illustrate BPS binds with StAR protein	[58]

## Data Availability

Not applicable.

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
