# Peer review of "Knowledge Gap in Understanding the Steroidogenic Acute Regulatory Protein Regulation in Steroidogenesis Following Exposure to Bisphenol A and Its Analogues"

_biomedicines, 2022, doi:10.3390/biomedicines10061281_

Round 1

Reviewer 1 Report

This paper reviews the StAR protein regulation, the cAMP-PKA, PLC-PKC, EGFR-MPAK/ERK and Ca2+-Nur77 signaling pathways for BPA and Its Analogues. It is mainly focused on testicular steroidogenesis in male reproductive system, and I think it will be of great help to those who study BPA and Its Analogues.
However, the title is too long, so I would like to make it concise. And BPA and Its Analogues have significant effects on ovarian follcles or granulosa cells in women. I would like to know if there are any plans for a review on it.

We present the parts that require English correction.
In this paper, Signalling --> Signaling
Line 14: Nowadays, the usage of bisphenols in our daily life has become extremely common. --> Nowadays, bisphenols in our daily lives have become extremely common. 
Line 16: StAR protein --> Steroidogenic acute regulatory protein (StAR) 
Line 17: which its regulation -->and its regulation
Line 21: MPAK/ERK --> MAPK/ERK 
Line 22: increased in ERK expression --> increased ERK expression 
Line 25: had found --> had been found 
Lines 26027: steroidogenesis is still 26 lacking which more research --> steroidogenesis is still lacking, and more research
Line 38: the children development --> the children's development
Line 40 development as well as changes --> development and changes 
Line 46: the use of BPA in --> BPA in 
Line 47: by FDA --> by the FDA
Line 52: other is the --> other are the 
Line 54: the manufacturer industry --> the manufacturing industry
Line 61: in the production of specialised --> in producing specialized 
Line 68: on organisms as BPA --> on organisms
Line 70: will give an impact --> will have an impact
Line 103: Transportation of --> The transportation of 
Line 105: StAR protein, which --> StAR protein which 
Line 110: in order to allow --> to allow 
Lines 127-129: Through a negative feedback loop, a high level of testosterone concentrations in the circulations triggers the hypothalamus-pituitary to suppress the LH secretion. --> A high level of testosterone concentrations in the circulations through a negative feedback loop triggers the hypothalamus-pituitary to suppress the LH secretion.  
Line 140: a number of transcription factors such as --> several transcription factors, such as     
Line 144: gions, appear --> gions appear 
Line 148: 1,4,5-trisphosphate (IP3) resulting --> 1,4,5-trisphosphate (IP3), resulting 
Line 158: in a number of MAPK-ERK --> in many MAPK-ERK   
Line 166: agents also involved --> agents are also involved 
Line 172: mobilisation --> mobilization
Line 175: by an increase --> by increasing 
Line 188: its receptor, LHR --> its receptor LHR 
Line 191: the LH either the LH --> the LH, either the LH 
Line 194: have shown that when adult male rats were exposed to multiple dosage of --> had shown that when adult male rats were exposed to multiple dosages of
Line 216: significant decreased in --> significant decrease in 
Line 218: be fully abolished by the lower of LHR --> be entirely abolished by the lower LHR
Line 223: BPA in in vitro studies --> BPA in vitro studies 
Lin 228: activate the AC. This eventually leads to inhibition of the cAMP pathway in such a way that BPA --> activating the AC. This eventually leads to inhibition of the cAMP pathway so that BPA
Line 235: study was found to report on the effect of BPA --> study reported the effect of BPA 
Line 236: Several previous studies are more focused --> Several previous studies focus 
Line 252: ability in binding to --> ability to bind to 
Lines 254-255: MAPK-ERK1/2 resulting --> MAPK-ERK1/2, resulting
Line 266: however, did not --> however, it did not 
Lin 269: reported in in vitro study --> reported in vitro study 
Line 271: and Akt resulting in an increase in Leydig cell --> and Akt in Leydig 
Line 272; A study by Ok et al. --> Ok et al. 
Line 273: ERK1/2 in in vitro --> ERK1/2 in vitro 
Line 275: was also found to be increased --> was also increased or was also found to increase 
Line 276: once in a week time --> once a week 
Line 294: Despite lack of --> Despite the lack of 
Line 296: placenta --> the placenta
Line 297: and resulted in --> and resulting in  
Line 299: remains --> remain
Line 307: StAR which --> StAR, which
Line 310: cascade which is the enzyme StAR --> cascade, the enzyme StAR
Line 320: overactivity, respectively in the Leydig cells --> overactivity, respectively, in the Leydig cells
Line 333: in 48 hours --> for 48 hours 
Line 337: BPA in in vitro study --> BPA in vitro study
Lines 338-339: Despite, their study showed that there is no involvement of--> Despite this, their study showed no involvement of  
Line 353: StAR either gene or protein expressions influence --> StAR, either gene or protein expressions, influences 
Line 356: decreased or no --> decreased, or no
Line 378: who was using --> using
Line 379: model for the assessment of BPA exposure on the differentiation of stem --> model to assess BPA exposure on the differentiation of stem cells 
Line 382: found in in vitro study --> found in vitro study
Line 389: to be increased in BPS --> to increase BPS 
Line 394: it is not involving --> it does not involve 
Line 395: ability in affecting StAR --> ability to affect StAR 
Lines 395-398: In silico docking studies exhibited that the ability of BPS, one of the BPA analogues, in binding with StAR protein may influence the transport of cholesterol into mitochondria which in turn modifies the steroidogenesis [58]. --> Silico docking studies exhibited that BPS, one of the BPA analogues, in binding with StAR protein may influence the transport of cholesterol into mitochondria which in turn modifies the steroidogenesis [58]. 
Lines 399-400: production, however, most of the studies stated that increase in StAR had defect the testicular steroidogenesis. --> production. However, most studies stated that an increase in StAR had the testicular steroidogenesis defect. 
Line 402: the cholesterol transportation --> cholesterol transportation
Line 403: affected resulting -->affected, resulting
Line 417: and decrease in --> and a decrease in 
Line 418: other hand, LHR expression was upregulated in the testis of male zebrafish which were --> other hands, LHR expression was upregulated in the testis of male zebrafish 
Line 419: Previous study --> A previous study 
Line 420: in contradict --> in contradictory 
Lins 423-424: This mechanism leads to reduce in StAR gene expression and reduced the testosterone synthesis. --> This mechanism reduces StAR gene expression and reduces the testosterone synthesis. 
Line 425: its analogues requires --> its analogues require
Line 431: Leydig cells is --> Leydig cells are
Line 433: analogues towards PLC-PKC --> analogues on PLC-PKC
Lines 433-435: Thus, this could be an opportunity for researchers to explore the effects of these EDCs towards this potential targeting protein, PKC in depth. --> Thus, this could be an opportunity for researchers to explore these EDCs' effects on this potential targeting protein, PKC, in-depth.
Line 441: and presumably, -->and, presumably,
Line 452: but involved in --> but is involved in 
Line 453-454: It is due to the increased concentration of Ca2+ could stimulate the formation of ROS in mitochondria during respiration leading to oxidative stress [84]. --> The increased concentration of Ca2+  could stimulate the formation of ROS in mitochondria during respiration leading to oxidative stress [84]. 
Line 455: Nur77 which --> Nur77, which 
Line 456: showed an increased --> showed an increase 
Line 458: which in turn results in the increase of the --> which in turn increases the 
Line 459: Normally, the StAR --> Usually, the StAR 
Line 461: StAR protein to be normally function in cholesterol mobilisation. --> StAR protein to function in cholesterol mobilization normally. 
Line 462: the elevated of Nur77 expression --> the elevated Nur77 expression 
Line 471: pathways which --> pathways, which
Line 482: circulation resulting in decreased of testosterone synthesis --> circulation results in decreased testosterone synthesis 
Lines 494-495: BPA and its analogues disturb the regulations of the StAR gene and protein via these three signalling pathways ultimately causing a negative impact on human health. --> BPA and its analogues disturb the StAR gene and protein regulations via these three signalling  pathways,  ultimately a negative impact on human health. 
Line 497: pathways has the possibility to downregulate --> pathways can downregulate 
Lines 499-501: Increased activation of the aromatase activity possibly may increase the oestradiol in the male reproductive system which ultimately increased the risk of prostate cancer. --> Increased activation of the aromatase activity may increase the oestradiol in the male reproductive system, which ultimately increases prostate cancer risk.
Line 503: approaches in ameliorating the adverse --> approaches in alleviating the adverse 

Reviewer 2 Report

General comments

The manuscript entitled “Lack of Evidence in Connecting the Signalling Pathways with Steroidogenic Acute Regulatory Protein Regulations in Testicular Steroidogenesis Exposed to BPA and Its Analogues: A MiniReview” gives a systematic overview of the effects of bisphenol A and its analogues on the StAR protein and signaling pathways involved in its regulations. 

It is a very nice, well-conducted, and written manuscript. This review clearly presented existing data and filled the gap in knowledge about the testicular toxicity of bisphenol A and its analogues. The manuscript is noteworthy and I would recommend it for publication.

Specific comments

Line 16: Please edit the sentence. StAR protein is not a precursorin steroidogenesis, StAR protein is rate-limiting step in steroidogenesis.

Table 1. The chemical structure of BPA and its analogues is not fully visible.

Line 349: Please put the full title: Effects of BPA and its analogues on StAR protein expression.

Reviewer 3 Report

Title should be changed. It is too long and confusing. As review shows impact of BPA on StAR protein and doesn't give insight in all known regulatory pathways it should be more focused on what is done. „Lack of evidences….“ In title i thus misleading.

Sentence „StAR protein is a precursor in steroidogenesis which its regulation involves several signalling pathway“ has no meaning. Rephrase.

Review either shows studies which support effect of BPA on StAR pathway molecules or shows that certain studies are not performed. Thus title and abstract are in coontradiction with presented facts. That means Abstract, part of Disucussion and Conclusion are in contradiction with presented body of evidences. Manuscript should have title which starts for example with „Knowledge gaps…….

As after all authors say „ we postulate that the BPA and its analogues affect the regulation of StAR in steroidogenesis via activation of cAMP-PKA, EGFR-MAPK/ERK and Ca2+-Nur77 signalling pathways which then ultimately decreased the StAR protein or its functionality.

Round 2

Reviewer 3 Report

Manuscript is improved following comments.
